# Dual-task walking reduces lower limb range of motion in individuals with Parkinson's disease and freezing of gait: But does it happen during what events through the gait cycle?

**Camila Pinto**[1,2�], **Ana Paula Salazar**[1,2☉], **Ewald Max Hennig**[3‡], **Graham Kerr**[3‡], **Aline Souza Pagnussat**[1,2‡]*

**1** Rehabilitation Sciences Graduate Program, Universidade Federal de Ciências da Saúde de Porto Alegre (UFCSPA), Porto Alegre, RS, Brazil, **2** Movement Analysis and Rehabilitation Laboratory, Universidade Federal de Ciências da Saúde de Porto Alegre (UFCSPA), Porto Alegre, RS, Brazil, **3** Institute of Health and Biomedical Innovation, Queensland University of Technology, Brisbane, QLD, Australia

☉ These authors contributed equally to this work.
‡ EMH, GK and ASP are the expert researchers and also contributed equally to this work.
* alinespagnussat@gmail.com

**Data Availability Statement:** All relevant data are within the manuscript and its Supporting Information files.

## Abstract

### Background

It is unclear how dual-task gait influences the lower limb range of motion (RoM) in people with Parkinson's disease (PD) and freezing of gait (FOG). The lower limb kinematics during dual-task gait might differ from regular gait, but during what events in the gait cycle?

### Methods

This is an observational within-subjects study. Thirty-two individuals with PD and FOG underwent a gait analysis. Single and dual-task gait was assessed by a 3D motion analysis system and the RoM data of the lower limb were extracted from hips, knees and ankles in the sagittal plane. Dual-task assignment was performed using word-color interference test. To compare both gait conditions, we used two different analyses: (1) common discrete analysis to provide lower limb RoM and (2) Statistical Parametric Mapping analysis (SPM) to provide lower limb joint kinematics. A correlation between lower limb RoM and spatiotemporal gait parameters was also performed for each gait condition.

### Results

Common discrete analysis evidenced reductions in RoM of hips, knees and ankles during the dual task gait when compared to single gait. SPM analysis showed reductions in flexion-extension of hip, knees and ankles joints when dual task was compared to single task gait. These reductions were observed in specific gait events as toe off (for knees and ankles) and heel strike (for all joints). The reduction in lower limb RoM was positively correlated with the reduction in step length and gait speed.

**Funding:** CP had her masters scholarship supported by Coordenação de Aperfeiçoamento de Pessoal de Nível Superior – Brasil (CAPES) – Finance Code 001.

**Competing interests:** The authors have declared that no competing interests exist.

## Conclusions

Lower limb joints kinematics were reduced during toe off and heel strike in dual task gait when compared to single gait. These findings might help physiotherapists to understand the influence of dual and single walking in lower limb RoM throughout the gait cycle in people with PD and FOG.

## Introduction

People with Parkinson's disease (PD) present motor deficits, which include bradykinesia, rigidity, postural and gait impairments [1, 2]. An important characteristic of PD is that motor symptoms are even worse when individuals need to perform two tasks simultaneously, e.g. dual-task gait [3, 4]. Walking while doing something else is very common in daily living, and it is a great challenge for people with PD [3, 5, 6]. As a result of the depletion of dopamine in basal ganglia circuits, individuals with PD rely more on the premotor cortex to achieve normal movement patterns. For this reason, they are affected by losses in the automaticity and rhythmicity of gait when cortical resources to complete a cognitive task are needed [7, 8].

Gait impairments during dual-task gait are even greater when individuals present other motor symptoms, such as the freezing of gait (FOG) [5, 9]. FOG affects over half of people with PD and is characterized by an absence or difficulty in stepping forward while walking (inability to lift feet from the floor) [5]. Freezers have impaired the perception of locomotor asymmetry, presenting a greater stride time variability across different walking conditions [10, 11]. Thus, dual-task gait may induce or aggravate FOG episodes and increase the risk of falls [3, 5, 6, 10].

Studies have reported spatiotemporal differences between single and dual-task gait [3, 12, 13]. When dual-task gait is compared to single task (regular walking), individuals with PD and FOG show slower speed, reduced step length and higher gait variability [3]. Further, gait performance worsens in the *off* phase of antiparkinsonian medication [2, 3]. Reductions in step length could be accompanied by a decrease in lower limb range of motion (RoM) [14, 15]. A previous study evaluated lower limb RoM (sagittal plane) in people with PD without FOG and control individuals during a single gait [16]. Individuals with PD showed a reduced RoM of hips, knees, and ankles when compared to healthy controls [16]. However, the correlation between spatiotemporal parameters and lower limb RoM and the effects of dual-task gait on lower limb RoM in people with FOG were not investigated yet.

Traditional gait analysis generally quantifies lower limb RoM as the average of the entire gait cycle through discrete parameter statistics. Through this methodology, it is a challenge to visualize movement trajectories into specific gait phases [2, 14, 17]. Statistical Parametric Mapping (SPM) is a statistical approach which allows non-direct hypothesis testing on kinematic data in a continuous way, considering the interdependence of the data points [17, 18]. SPM is a method able to analyze one-dimensional (1D) data and identify the exact pattern and location of joint kinematics during the entire gait cycle [17–19]. SPM may be used as complementary to discrete analysis to detect periods with significant between-group differences through the total cycle of movement [17, 18]. In this sense, this analysis reduces the risk of Type I error providing a detailed way to visualize the kinematics throughout an specific task (i.e. gait) [20].

Therefore, this study aimed: (1) compare lower extremities RoM during single and dual-task gait (by means of discrete analysis); (2) compare lower limb joint kinematics during single and dual-task gait at each point across the gait cycle (by means of SPM) in individuals with PD and FOG during partially *off* state of antiparkinsonian medication. Also, we aimed to verify

the association between lower limb RoM and spatiotemporal parameters, as gait speed and step length.

## Methods

### Participants

This observational within-subjects study followed the "Strengthening the Reporting of Observational Studies in Epidemiology" (STROBE) [21] checklist and was approved by the Ethics Committee of the Universidade Federal de Ciências da Saúde de Porto Alegre (protocol 1.333.131). All participants signed the informed consent before starting the procedures.

Inclusion criteria were diagnosis of idiopathic PD (according to the London Brain Bank Criteria) [22], age between 50 and 85 years (in order to encompass most of PD population) [23], capacity of walking at least eight meters unassisted or with assistance, regular FOG episodes (verified by the Freezing of Gait Questionnaire—FOG-Q) and minimum score of 20 in the Mini Mental State Examination (MMSE). Exclusion criteria were the presence of deep brain stimulation devices, peripheral neuropathy and musculoskeletal or neurological problems that impaired gait.

We chose to evaluate participants in the *off* phase in order to avoid dopamine effects on FOG [24]. Participants were evaluated in the end-of dose of medication (partially or close to *off* state), when levodopa is losing its effect ("wearing *off*"). Most of the participants did not tolerate withdrawal from levodopa medication for 12 hours in their daily living. For this reason, the *off*-medication phase was defined according to the end-of dose of medication as the participant's drug regimen in order to reproduce their daily routine where medication is losing its effect. Individuals were instructed to not intake the next dose at the evaluation time, according to each prescribed time of medication intake (it ranged between 4 to 12 hours, according to each participant). If the researchers noticed that individuals were still in "*on*-phase", they waited until a subjective "*off* state" to start the tests.

### Measures

All procedures were conducted at the Movement Analysis and Rehabilitation Laboratory at the Universidade Federal de Ciências da Saúde de Porto Alegre. Unified Parkinson's Disease Rating Scale (UPDRS III) and Modified Hoehn & Yahr scale (H&Y) were used to classify the sample regarding motor ability and severity of disease in the *off* and *on* medication stage [25]. Gait data were acquired using a 3D motion analysis system (BTS SMART DX 400 Motion Capture System, Milan, Italy). After being instructed about the procedures, a researcher with experience in motion analysis placed 22 reflective spherical markers (15 mm diameter) on the participant's body, as described in the Davis protocol [26]. Thus, participants were instructed to perform regular walking (single task gait) at their self-selected velocity in a path of 8 m x 1.4 m (their walking speed were not controlled). Then, they were asked to walk while performing a cognitive test at the same time (dual-task gait). The cognitive test performed was the word-color interference test [27] which consists of reading aloud the name of colors written in non-congruent colors. Participants walked at least six times in each condition (single and dual-task gait). They first completed regular walking trials and, then, all dual-task trials. They were all welcome to rest if they needed to along the trials to avoid fatigue. Three participants used a walking stick during the evaluation and two presented FOG episodes during gait evaluation.

The first walking trial was used for familiarization; therefore, it was not analyzed. At least, five gait cycles were analyzed for each participant. Trials were temporally normalized to 100% for a complete gait cycle and the raw data were processed using the BTS Smart Analyzer software. Joint ranges in sagittal plane and spatiotemporal data were calculated for at least three

strides which provided RoM values from right and left hip flexion-extension (Hip-FE), knee flexion-extension (Knee-FE), and ankle plantar-dorsiflexion (Ankle-PD). The average of three randomly selected trials was extracted for each gait condition to be included in the final statistics. Those data were analyzed by discrete analysis to compare lower extremities RoM between single and dual-task gait. Also, the gait speed change between single and dual-task gait was calculated using Dual Task Effect (DTE) formula [DTE (%) = (dual task gait speed–single task gait speed) / single task gait speed x 100%] [30]. Negative speed DTE values indicate a decrement under dual compared to single task.

SPM analysis was also conducted to analyze the joint kinematics in sagittal plane across an entire waveform taking account the interdependence of datapoints (100 for a whole gait cycle). For this purpose, the kinematic raw data were extracted, filtered and analyzed using a custom Matlab program (The Mathworks Inc, Natick, MA) to perform SPM analysis. SPM shows the joint kinematics at each point across a gait cycle (100%). In order to better translate data for clinical practice, we matched each percentage of the gait cycle based on well-established gait events, as follows: initial contact or heel strike (approx. 0%), load response (approx. 10%), heel off (approx. 30%), opposite initial contact (approx. 50%), toe off (approx. 60%), feet adjacent (approx. 73%), tibial vertical (approx. 87%), and next initial contact or heel strike (approx. 100%) [28].

## Statistical analysis

Twenty-six participants were calculated as necessary to detect a mean difference of 3.59 degrees in the knee RoM, with standard deviation of 5.8, 80% of power and alpha value of 0.05 [8]. We used the software G Power 3.0.10 for sample size calculation. Data normality was tested using Shapiro-Wilk tests and the homogeneity of variance was tested by Levene's statistic. Paired sample t-tests were used for common discrete analysis to assess differences between single and dual-task gait—average of lower limb RoM and spatiotemporal parameters (step length and velocity).

Correlations were also performed using Pearson's correlation. Firstly, single versus dual-task gait was compared regarding lower limb RoM. Secondly, spatiotemporal parameters versus lower limb RoM were compared during single gait and dual-task gait separately. Correlation values were considered very high (0.90 to 1.00), high (0.70 to 0.90), moderate (0.50 to 0.70), low (0.30 to 0.50), or negligible (0.00 to 0.30) [29]. Multiple Linear Regression Analyses were performed using the block-wise selection to determine if gait speed (independent variable) would be a predictor to explain the variance on the RoM of lower limbs (dependent variables) for each condition separately (single and dual-gait). SPSS® Statistics 20.0 (Chicago, IL, USA) was used for analyses.

To compare gait conditions, a custom Matlab program (The Mathworks Inc, Natick, MA) was used to conduct 1D SPM analysis using the open-source spm1d code (version 0.4, http://www.spm1d.org) [30] as described in previous study [18]. SPM1D uses the single inference method to calculate significance of temporal clusters, or regions of next values for which the statistic test exceeds the significance threshold (supra-threshold clusters). Gait trials of all participants were filtered and included in the analysis with the same number of datapoints. For this method, a single p-value is reported for each observed cluster above the threshold. Paired sample t-tests were used to compare kinematic data of hip, knee and ankle joint kinematics during single task and dual-task gait. All results were considered statistically significant for p<0.05.

## Results

Thirty-two people with PD and FOG were included. Only one participant was excluded from the statistical analysis because data were corrupted. Table 1 depicts the characteristics of participants.

**Table 1. Demographic and clinical characteristics.**

|  | Individuals with PD and FOG (n = 32) |
|---|---|
| Sex (F/M) | 9/23 |
| Age (years) | 65.13 (61.78, 68.47) |
| Body Mass (kg) | 76.72 (68.57, 83.87) |
| Height (cm) | 164 (160, 168) |
| Time of disease (years) | 9.19 (7.35, 11.04) |
| MMSE | 26.55 (25.24, 27.86) |
| FOG-Q | 13.97 (12.20, 15.74) |
| H&Y (*off*) |  |
| 1 / 1.5 / 2 / 2.5 | 1 / 1 / 4 / 6 |
| 3 / 4 | 11 / 8 |
| 5 | 1 |
| H&Y (*on*) |  |
| 1 / 1.5 / 2 / 2.5 | 6 / 5 / 4 / 9 |
| 3 / 4 | 5 / 3 |
| 5 | 0 |
| UPDRS III (*off*) | 23.97 (20.83, 27.11) |
| UPDRS III (*on*) | 11.00 (11.63, 15.79) |
| LEDD (mg) | 1044 (867, 1232) |

*Note*. Data are mean and 95% confidence intervals (lower bound, upper bound). H&Y and sex are in frequencies.
Abbreviations: PD: Parkinson disease; FOG: Freezing of gait; MMSE: Mini mental state examination; FOG-Q:
Freezing of gait questionnaire; H&Y (*off*): Modified Hoehn & Yahr scale during *off*-medication state; UPDRS III
(*off*): Motor part of the Unified Parkinson's disease rating scale during *off*-medication state; LEDD: Levodopa
equivalent daily medication dosage [31].

Common discrete analysis was used to assess the average of lower limb joint angles in the
sagittal plane during single and dual-task gait. RoM of hips, knees, and ankles in the sagittal
plane were significantly lower during the dual-task gait (p<0.01) (Table 2). There were no differences in the lower limb RoM between right and left sides.

**Table 2. Joint angles and spatiotemporal parameters during gait (single task and dual task) in individuals with PD and FOG.**

|  | Single Task | Dual Task | Paired Differences | p-value |
|---|---|---|---|---|
| R Hip-FE (˚) | 35.08 (31.92, 38.24) | 29.76 (26.38, 33.13)* | **-5.32 (-7.57, -3.07)** | **.001** |
| L Hip-FE (˚) | 33.68 (30.36, 37.01) | 29.05 (24.81, 33.28)* | **-4.63 (-8.12, -1.13)** | **.011** |
| R Knee-FE (˚) | 46.58 (42.78, 50.37) | 41.75 (37.17, 46.34)* | **-4.82 (-7.14, -2.50)** | **.001** |
| L Knee-FE (˚) | 46.89 (43.47, 50.31) | 41.32 (36.60, 46.05)* | **-5.56 (-9.07, -2.50)** | **.003** |
| R Ankle-PD (˚) | 27.03 (23.60, 30.46) | 23.51 (20.11, 26.90)* | **-4.81 (-9.07, -2.49)** | **.001** |
| L Ankle-PD (˚) | 27.94 (24.27, 31.61) | 24.62 (20.52, 28.72)* | **-3.31 (-5.12, -1.51)** | **.001** |
| R Step Length (cm) | 0.43 (0.36, 0.48) | 0.33 (0.27, 0.38)* | **-0.12 (-0.13, -0.05)** | **.001** |
| L Step Length (cm) | 0.45 (0.39, 0.50) | 0.32 (0.25, 0.38)* | **-0.09 (-0.17, -0.08)** | **.001** |
| Speed (m/s) | 0.79 (0.69, 0.89) | 0.58 (0.48, 0.68)* | **-0.21 (-0.28, -0.14)** | **.001** |
| Speed DTE (%) | - | -40 (-80.74, -32.12) | - | - |

*Note*. Data are mean and 95% confidence intervals (lower bound, upper bound). Joint angles are in degrees (˚).
Abbreviations: PD: Parkinson's disease; R: Right; L: Left; Hip-FE: Hip flexion/extension; Knee-FE: Knee flexion/extension; Ankle-PD: Ankle plantar/dorsiflexion, DTE:
Dual task effect.
* significant difference between single task and dual task.

All lower limb RoM were positively correlated with spatiotemporal gait parameters (step length and gait speed) in single and dual-task gait respectively (p<0.02). Hip and knee RoM were strongly correlated with step length and gait speed, although ankle presented a moderate correlation. From the multiple regression analyses, it was possible to predict the performance of lower limb RoM from a potential predictor: gait speed. In this matter, we used the right side of lower limb RoM to perform the regression analysis as we didn't find differences between the right and left sides. Gait speed during single gait explained 70.8% of the variance of hip RoM, 68.2% of knee, and 28.0% of ankle RoM. On the other hand, gait speed during dual-task gait explained 49.9% of hip RoM, 61.2% of knee, and 36.8% of ankle RoM. Table 3 presents correlations between lower limb RoM and spatiotemporal gait parameters and regression analyses between lower limb RoM and gait speed.

SPM analysis evidenced joint reductions on sagittal plane during dual-task gait in left hip, both knees and ankles in comparison with single task. Flexion-extension was reduced in left hip when feet were adjacent (left hip: 76–100%, Tcritical = 2.952, p = 0.003) (Fig 1). Flexion-extension was also reduced in both knees when toe was off the ground to feet adjacent (right knee: 58–70%, p = 0.002, Tcritical: 3.171; left knee: 59–73%, p = 0.005, Tcritical = 3.135) and near to the next heel strike (right knee: 89–99%, p = 0.008, Tcritical = 3.171, left knee: 90–100%, p = 0.009, Tcritical = 3.135) (Fig 2). Flexion-extension was reduced in both ankles when toe was off the ground (right ankle: 59–68%, p = 0.007, Tcritical: 3.236; left ankle: 59–68%,

**Table 3. Correlations and linear multiple regression between joint angles and spatiotemporal parameters during gait (single task and dual task) in individuals with PD and FOG.**

| | | R Hip-FE(˚) | L Hip-FE(˚) | R Knee-FE(˚) | L Knee-FE(˚) | R Ankle-PD(˚) | L Ankle-PD(˚) |
|---|---|---|---|---|---|---|---|
| **Single Task** | | | | | | | |
| **Correlations** | | | | | | | |
| R Step Length (cm) | Correlation Coefficient | .892 | .718 | .843 | .501 | .616 | .484 |
| | p-value | < .001 | < .001 | < .001 | .003 | < .001 | .005 |
| L Step Length (cm) | Correlation Coefficient | .852 | .759 | .839 | .574 | .525 | .389 |
| | p-value | < .001 | < .001 | < .001 | .001 | < .001 | .028 |
| Speed (m/s) | Correlation Coefficient | .847 | .742 | .832 | .595 | .537 | .392 |
| | p-value | < .001 | < .001 | < .001 | < .001 | < .001 | .027 |
| **Linear Multiple Regression** | | | | | | | |
| Speed (m/s) | Adjusted R Square | .708 | - | .682 | - | .280 | - |
| | p-value | < .001 | - | < .001 | - | < .001 | - |
| **Dual Task** | | | | | | | |
| **Correlations** | | | | | | | |
| R Step Length (cm) | Correlation Coefficient | .819 | .652 | .787 | .735 | .618 | .507 |
| | p-value | < .001 | < .001 | < .001 | < .001 | < .001 | .003 |
| L Step Length (cm) | Correlation Coefficient | .839 | .798 | .814 | .854 | .645 | .628 |
| | p-value | < .001 | < .001 | < .001 | < .001 | < .001 | < .001 |
| Speed (m/s) | Correlation Coefficient | .718 | .726 | .791 | .789 | .624 | .595 |
| | p-value | < .001 | < .001 | < .001 | < .001 | < .001 | < .001 |
| **Linear Multiple Regression** | | | | | | | |
| Speed (m/s) | Adjusted R Square | .499 | - | .612 | - | .368 | - |
| | p-value | < .001 | - | < .001 | - | < .001 | - |

*Note*. Joint angles are in degrees (˚). The joint angles of right side were used for the linear multiple regression analysis as there were no differences between the right and left sides.

Abbreviations: R: Right; L: Left; Hip-FE: Hip flexion/extension; Knee-FE: Knee flexion/extension; Ankle-PD: Ankle plantar/dorsiflexion.

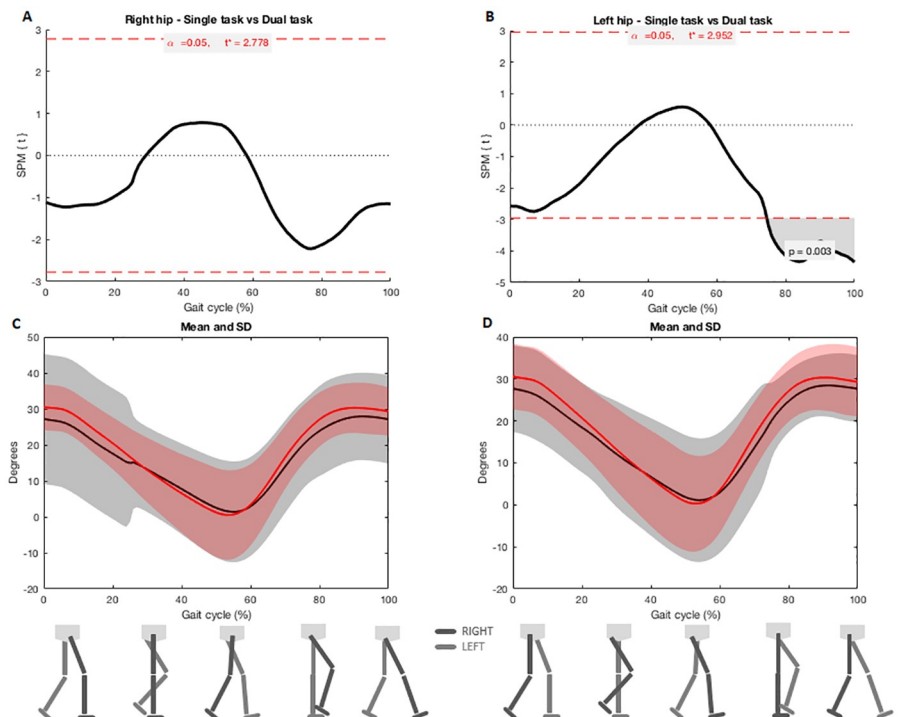

**Fig 1.** A and B) SPM analysis of hip angles (right and left) during single and dual-task gait; C and D) Joint angles during the entire cycle of gait. Black lines indicate angles during the dual-task gait and red lines indicate single task gait. Shaded areas represent the standard deviation.

p = 0.007, Tcritical = 3.207) and near to the next heel strike (right ankle: 91–95%, p = 0.027, Tcritical: 3.236) (Fig 3).

## Discussion

This study showed the exact gait phases in which dual-task gait interfered on the joint kinematics of lower limb when compared to single gait in people with PD and FOG. Most of daily activities include dual task situations, which require an appropriate body response [8], especially for people with PD. The knowledge of how this challenge task influence lower limb kinematics may be important for gait training strategies in this population. To this intent, the lower limb patterns during single and dual-task gait were evaluated using two different statistical analyses: (1) common discrete analysis and (2) SPM analysis. Both statistical approaches have distinct purposes. The first one aimed to compare the RoM of lower limb although the second analysis accessed the performance of lower limb kinematics at each point throughout both walking cycles. When compared to time series analysis, SPM is considered a suitable method to analyze 1D data and a consistent way to interpret clinical findings [32].

The RoM of all analyzed joints (hips, knees and, ankles) was reduced during dual-task walking compared to single task walking using discrete analysis. Perhaps, as an attempt to seek for stability during the dual-task gait due the double demand on cortex, individuals with PD and FOG may diminish the RoM of lower limb. One previous study compared lower extremity RoM of individuals with PD during single and dual-task gait during the *on* state of medication [8]. Authors did not find any differences on hips, knees and ankles RoM comparing both conditions, although values for dual-task were slightly lower than those from single gait [8].

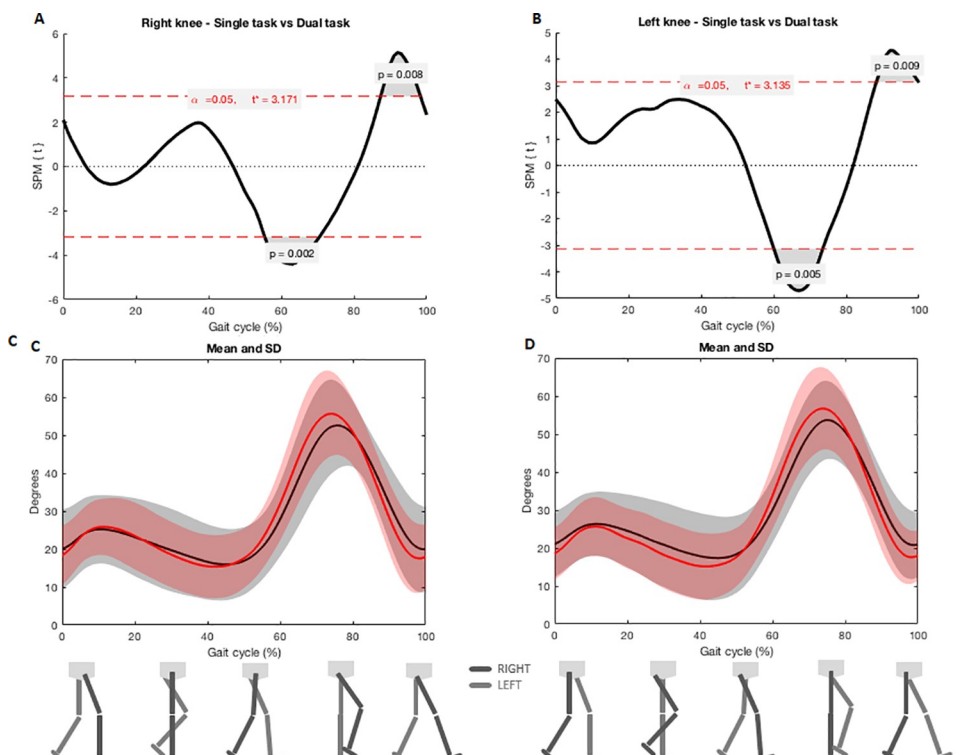

**Fig 2.** A and B) SPM analysis of knee angles (right and left) during single and dual-task gait. Gray zones indicate the exact moment of the gait cycle in which angles differ; C and D) Joint angles during the entire cycle of gait. Black lines indicate angles during the dual-task gait and red lines indicate single task gait. Shaded areas represent the standard deviation.

Distinctly, our study had some differences. For instance, we included only individuals with FOG, most of them in moderate or severe stages (3 and 4 respectively) according to H&Y scale. Additionally, participants performed their walking trials close to their *off* antiparkinsonian medication phase (partially *off*) because this condition is more common to happen during their daily living and may influence on FOG [2, 8]. In this sense, we hypothesized that FOG and *off* state may influence on diminishing lower extremities RoM in dual gait situations. However, this study did not address this question about which of situations (i.e. FOG or *off* state of medication) had a pronounced interference.

This study also aimed to understand in which gait event or period lower limb joint kinematics are more affected. For this purpose, we used SPM analysis. We observed greater left hip extension during the mid to terminal swing period (approx. 76–100%) in dual-task gait when compared with regular gait. In this period it is necessary to have a maximal hip flexion of 30 degrees before the heel contact [33]. SPM analysis showed that participants ended the next heel strike event or terminal swing period (approx. 89–100%) with their knees more flexed during dual task when compared to single task gait. The last period of swing requires complete knee extension to prepare for stance and allows ankle dorsiflexion during the heel strike in the initial contact [34]. Thus, the lack of knee extension could prevent the required ankle movement in the following gait phase. At the start of the swing period, which is the moment of withdrawal of the toes (approx. 60% of the gait cycle), the knee flexion begins and reaches approximately 35 degrees of flexion to a maximum flexion of approximately 60 degrees (approx. 73% of the gait cycle) [33]. During dual-task gait, knees were more extended during

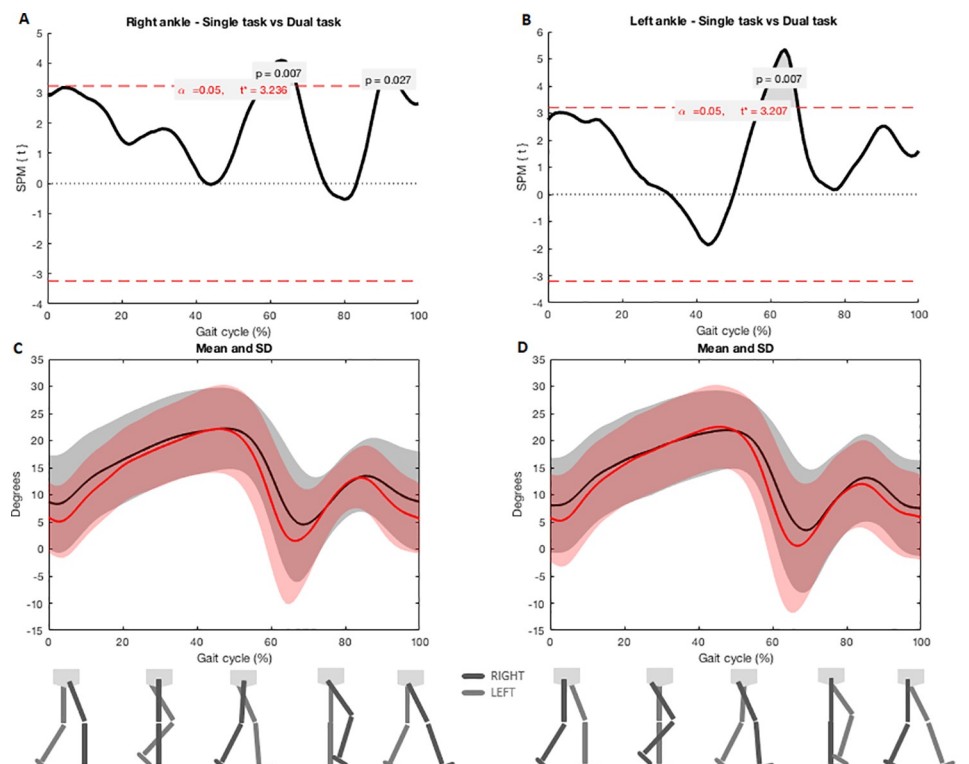

**Fig 3.** A e B) SPM analysis of ankle angles (right and left) during single and dual-task gait. Gray zones indicate the exact moment of the gait cycle in which angles differ; C and D) Joint angles during the entire cycle of gait. Black lines indicate angles during the dual-task gait and red lines indicate single task gait. Shaded areas represent the standard deviation.

the toe off event during initial swing period. The lack of knee flexion during the swing phase interferes with toe withdrawal and forward progression [33].

Kinematic waveforms of ankles dorsi-plantarflexion were reduced into two key points of the dual-task gait cycle: (a) pre and initial swing periods (approx. 59–68%), and (b) terminal swing period (approx. 91–95%). A slight plantarflexion in the loading response is critical for the heel rocker initiation of limb progression (which was reduced during dual-task gait, although not significant). A reduced ankle plantarflexion could result in a reduced propulsion for the next swing phase and, possibly, in a shorter step length [33]. Pre and initial swing periods occur when the toe rises and swings [33]. During the pre-swing, ankle activity is more related to the progression than to the weight bearing. In the initial swing, ankle mobility for plantarflexion is also necessary for limb advancement [34]. For individuals with PD, flexing the limb for floor clearance in the swing phase is a challenging part of the cycle [14, 15]. In general, participants maintained their ankles in constant dorsiflexion and failed to perform plantarflexion. The toe off event needs about 20 degrees of ankle plantarflexion to allow the propulsion needed for limb advancement and avoid trips and falls. According to our results, there were reductions in lower limb kinematics during dual-task gait when compared to single gait, particularly during toe off and heel strike.

Other studies have hypothesized that the reduced step length during dual-task walking could be related to a multijoint reduction in amplitude [2]. Our results confirm this assumption for the first time. We found significantly correlations between lower limb RoM and spatiotemporal gait parameters (step length and gait speed). It means that shorter steps length and

gait slowness are highly associated to a multijoint reduction in lower limb amplitude [35]. Particularly, hips and knees might suffer more influence of step length and gait speed as they presented a strong positively correlation. The reduction in lower limb RoM may be related to decrements in step length to anticipate/increase the double support phase. Thus, modifications in lower limb RoM could also have an influence on stability during the weight bearing, foot clearance, limb advancement and progression. Moreover, previous studies discuss an important influence of gait speed on lower limb kinematics [36], concluding that gait speed is angle-dependent and dependent on the phase of the gait cycle [37]. It means that the RoM of lower limb joints increases with a faster walk [36]. In order to assess the contribution of gait speed on lower limb RoM, we performed a regression analysis. It seems that slower or faster walking has a strong relationship with how much individuals increase the RoM of lower limbs, especially for hips and knees. We hypothesize that when individuals with PD focus on another activity (i.e. dual-task gait), they reduce both the gait speed and the RoM of lower limbs in order to increase stability.

There are limitations that must be considered when reading this study. For this reason, the generalization of our results should be carried out with caution. An important point is that only two individuals presented FOG episodes during the tests. Although all participants presented FOG symptoms according FOG-Q and dual gait could provoke these episodes, the laboratory environment may have an influence on participant attention during the evaluations. Thus, it is hard to know precisely the contribution of FOG on our results. However, freezers are more likely to present a worse gait performance when compared to non-freezers [11] and perhaps it still remains even walking without FOG episodes. The non-inclusion of people without FOG and paired healthy controls should be considered another limitation. In addition, participants were not in the optimal *off* state of medication (12 hours of withdraw) and participant's performance can have had a slight levodopa influence. On the other hand, situations when medication is losing its effect (wearing or partially off state) may be more common during patient's daily routine than those with 12 hours of medication withdraw. Another important limitation is that although we demonstrated changes in lower limb kinematics during dual-task gait when compared to single gait, particularly during toe off and heel strike, reductions in gait speed were also observed in the dual-task condition, which raises the potential for confounding. It is unknown whether these changes in lower limb kinematics result directly and uniquely from the dual-task condition, or whether similar changes would be observed in a single-task condition with slower gait speed. To find out how much lower limb kinematics are speed related, individuals should walk at matched speeds in dual and single task conditions in futures studies.

Therefore, this study aimed to supply a more detailed understanding of lower limb kinematics during single and dual-task gait in individuals with PD and FOG. Dual-task gait reduced the RoM of hips, knees, and ankles when compared to single gait. Also, step length and gait speed showed a strong influence on lower limb RoM during both gait conditions. Particularly, reduced flexion-extension of lower limb kinematics during dual task gait occurred during toe off and heel strike events. Knowing exactly where the change on lower limb kinematics occurs could help physiotherapists to focus on specific gait events and improve rehabilitation processes. Further studies are necessary to analyze if regaining joint mobility in those gait events would have an impact on gait performance of individuals with PD with and without FOG.

## Supporting information

**S1 Appendix. Data availability for discrete analysis (SPSS) and SPM analysis (Matlab) for each joint and gait condition.** *Note.* R: right; L: left; DT: dual task; ST: single task. (XLSX)

## Acknowledgments

We thank all the participants for their time and cooperation.

## Author Contributions

**Conceptualization:** Camila Pinto, Ana Paula Salazar, Aline Souza Pagnussat.

**Data curation:** Camila Pinto, Ana Paula Salazar.

**Formal analysis:** Camila Pinto, Ana Paula Salazar.

**Investigation:** Camila Pinto, Aline Souza Pagnussat.

**Methodology:** Camila Pinto.

**Project administration:** Camila Pinto, Ana Paula Salazar, Ewald Max Hennig, Aline Souza Pagnussat.

**Supervision:** Aline Souza Pagnussat.

**Validation:** Aline Souza Pagnussat.

**Visualization:** Graham Kerr.

**Writing – original draft:** Camila Pinto.

**Writing – review & editing:** Camila Pinto, Ana Paula Salazar, Ewald Max Hennig, Graham Kerr, Aline Souza Pagnussat.

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
