## [Decision Letter · Decision Letter 0]

4 May 2020

PONE-D-20-03253

Dual-task walking reduces lower limb range of motion in individuals with Parkinson’s Disease and Freezing of Gait. But during which phase of gait does it happen?

PLOS ONE

Dear Dr. Pagnussat,

Thank you for submitting your manuscript to PLOS ONE. After careful consideration, we feel that it has merit but does not fully meet PLOS ONE’s publication criteria as it currently stands. Therefore, we invite you to submit a revised version of the manuscript that addresses the points raised during the review process.

We would appreciate receiving your revised manuscript by Jun 18 2020 11:59PM. To enhance the reproducibility of your results, we recommend that if applicable you deposit your laboratory protocols in protocols.io, where a protocol can be assigned its own identifier (DOI) such that it can be cited independently in the future. For instructions see: http://journals.plos.org/plosone/s/submission-guidelines#loc-laboratory-protocols

We look forward to receiving your revised manuscript.

Kind regards,

J. Lucas McKay, Ph.D., M.S.C.R.

Academic Editor

PLOS ONE

Journal Requirements:

Reviewers' comments:

Reviewer's Responses to Questions

**Comments to the Author**

1. Is the manuscript technically sound, and do the data support the conclusions?

Reviewer #1: Partly

2. Has the statistical analysis been performed appropriately and rigorously? 

Reviewer #1: I Don't Know

3. Have the authors made all data underlying the findings in their manuscript fully available?

Reviewer #1: No

4. Is the manuscript presented in an intelligible fashion and written in standard English?

Reviewer #1: Yes

5. Review Comments to the Author

Reviewer #1: This manuscript investigates the range of motion (ROM) of sagittal plane joint angles during dual-task gait in individuals with Parkinson’s disease (PD) that have freezing of gait (FOG). In a cross-section study of 32 individuals with PD-FOG, the authors found that the range-of-motion was reduced in most sagittal plane angles in the dual-task condition. Using a statistical parametric mapping (SPM) analysis, the authors further found that these reductions occurred during specific phases of the gait cycle: pre-swing into swing phases. Although this study represents a large experimental effort with many subjects that this reviewer recognizes are likely hard to recruit, there are several methodological issues that must be addressed before the conclusions can be supported.

1. SPM does not identify ROM. The purpose of this study was to investigate the differences in ROM between dual-task and single-task gait conditions. As the authors discuss, the most common method to calculate ROM is discrete analysis over the entire gait cycle but this analysis cannot identify sub-phases of the gait cycle in which deviations in ROM occur. The authors supplement discrete analysis with SPM, however SPM does not identify ROM. ROM is the range over which a joint moves. SPM analysis identifies differences in joint kinematics at each point across the gait cycle, i.e., not a range. Certainly, the application of SPM to analyze gait can provide a more detailed understanding of gait kinematics compared to traditional discrete ROM analysis. I suggest the authors re-write this manuscript with this important distinction

2. Related to the above point, the comparison between SPM and discrete analysis does not seem fair. Why is the ROM only calculated over the entire gait cycle? A fairer comparison would be to also calculate ROM using discrete analysis within each of the gait phases. Likely this will not result in many significant differences, since it is not powered for such repeated measures. A lack of difference in the discrete sub-phase ROM would provide more support for the utility of SPM.

3. The methodology is not clearly written and is lacking many details. For example:

a. Lines 148-151 list the phases of gait that were analyzed, but it is not very clear what is done with the gait phases. Since this information is listed before any description of the discrete and SPM analyses, this reviewer expected that both methodologies would examine ROM during each gait phase (see point 2 above regarding the importance of that analysis).

b. There is minimal detail on the ROM and SPM methodology. How was ROM calculated? How does SPM work? Etc.

c. The authors analyzed the correlation between ROM and gait parameters. This analysis is not clearly described. Was this analysis done on ALL values (single and dual task conditions combined) or only one of the conditions? It is also not clear why this correlation was not compared between the dual-task and single-task conditions.

4. The limitations paragraph (last full paragraph on page 14) suggests a lot of limitations of the current study, including lack of control groups as well as FOG and OFF-med related limitations. These are all important limitations, yet the authors have not discussed what these limitations mean in the context of this study and the results. How much the results be affected by not capturing FOG episodes, for example? Do the authors expect that individuals with FOG subtype of PD have altered gait even without FOG episodes or are these differences representative of PD in general? Similarly, how do authors expect that being in the non-optimal OFF state may affect the results? Having appropriate control groups is critical to answering these questions. This is listed as a limitation, but this limitation is much bigger than it was given credit for. Given the likely difficulty to collect more data with many countries being shut down to due COVID-19, this reviewer suggests at the minimum to compare the current results with results in existing literature. Without this comparison, the results that PD-FOG results in the presented ROM differences is not supported.

Minor Comments:

Methods, Line 105: Why was the age range for inclusion criteria 50-85 years? There is no rationale give for this.

Methods, Lines 111-119: What is the rationale for testing in the OFF state? Or rather, close to OFF state as possible. This is in the Discussion section, but not motivated in either the Intro or Discussion.

Methods, Lines 148-151: Why were gait phases based on percentages instead of gait events? How consistent were these percentages across subjects (with gait events based on marker data)?

Methods / Results, Table 2: Only DTE on the gait task was evaluated. What was the Cognitive DTE.

6. PLOS authors have the option to publish the peer review history of their article (what does this mean?). If published, this will include your full peer review and any attached files.

Reviewer #1: No

---

## [Author Response · Author response to Decision Letter 0]

22 Jun 2020

Review Comments to the Author

Reviewer #1: This manuscript investigates the range of motion (ROM) of sagittal plane joint angles during dual-task gait in individuals with Parkinson’s disease (PD) that have freezing of gait (FOG). In a cross-section study of 32 individuals with PD-FOG, the authors found that the range-of-motion was reduced in most sagittal plane angles in the dual-task condition. Using a statistical parametric mapping (SPM) analysis, the authors further found that these reductions occurred during specific phases of the gait cycle: pre-swing into swing phases. Although this study represents a large experimental effort with many subjects that this reviewer recognizes are likely hard to recruit, there are several methodological issues that must be addressed before the conclusions can be supported.

1. SPM does not identify ROM. The purpose of this study was to investigate the differences in ROM between dual-task and single-task gait conditions. As the authors discuss, the most common method to calculate ROM is discrete analysis over the entire gait cycle but this analysis cannot identify sub-phases of the gait cycle in which deviations in ROM occur. The authors supplement discrete analysis with SPM, however SPM does not identify ROM. ROM is the range over which a joint moves. SPM analysis identifies differences in joint kinematics at each point across the gait cycle, i.e., not a range. Certainly, the application of SPM to analyze gait can provide a more detailed understanding of gait kinematics compared to traditional discrete ROM analysis. I suggest the authors re-write this manuscript with this important distinction

Response: We strongly agree with the reviewer and we apologize this misunderstanding. Our study aimed to compare lower limbs ROM between single and dual-task gait in people with PD and FOG but it also aimed to supply it with joint kinematics behavior in each point during the locomotion. We understand the important distinction between both discrete and SPM analysis and we made several modifications across the whole manuscript to address this question. 

2. Related to the above point, the comparison between SPM and discrete analysis does not seem fair. Why is the ROM only calculated over the entire gait cycle? A fairer comparison would be to also calculate ROM using discrete analysis within each of the gait phases. Likely this will not result in many significant differences, since it is not powered for such repeated measures. A lack of difference in the discrete sub-phase ROM would provide more support for the utility of SPM.

Response: The reviewer made a good point. The comparison between SPM and time series analysis concluded that SPM is a suitable method to analyze 1D data and could reduce the risk of Type I error providing a detailed way to visualize the kinematics throughout a specific task. On the other hand, SPM analysis might not replace discrete analysis, but it must be used as a complementary approach to discrete analysis (Northeast L, et al. 2018). We carefully reread all the manuscript and made several modifications in this sense across the introduction (pages 3 and 4) and discussion (page 17).

Northeast L, Gautrey CN, Bottoms L, Hughes G, Mitchell ACS, et al. (2018) Full gait cycle analysis of lower limb and trunk kinematics and muscle activations during walking in participants with and without ankle instability. Gait Posture 64: 114-118.

3. The methodology is not clearly written and is lacking many details. For example:

a. Lines 148-151 list the phases of gait that were analyzed, but it is not very clear what is done with the gait phases. Since this information is listed before any description of the discrete and SPM analyses, this reviewer expected that both methodologies would examine ROM during each gait phase (see point 2 above regarding the importance of that analysis).

Response: As requested previously, we clarified the important distinction between both discrete and SPM analysis through the manuscript as well as replaced gait phases by gait events (pages 6-8).

b. There is minimal detail on the ROM and SPM methodology. How was ROM calculated? How does SPM work? Etc.

Response: To address this question, we rewrote the methodology to detail the analysis (pages 6-8).

c. The authors analyzed the correlation between ROM and gait parameters. This analysis is not clearly described. Was this analysis done on ALL values (single and dual task conditions combined) or only one of the conditions? It is also not clear why this correlation was not compared between the dual-task and single-task conditions.

Response: We performed a correlation analysis between spatiotemporal parameters and lower limb ROM as an additional analysis. We aimed to explore if spatiotemporal parameters were related to lower limb ROM and we found a strong correlation between these parameters. We detailed these correlations across the document (methods, results and discussion sessions). Regarding the comparison between dual and single conditions, we already expected that these conditions would be correlated, and the statistical analysis confirmed that. We decided to not include this extra information in the paper to avoid a misunderstanding of the reader. 

4. The limitations paragraph (last full paragraph on page 14) suggests a lot of limitations of the current study, including lack of control groups as well as FOG and OFF-med related limitations. These are all important limitations, yet the authors have not discussed what these limitations mean in the context of this study and the results. How much the results be affected by not capturing FOG episodes, for example? Do the authors expect that individuals with FOG subtype of PD have altered gait even without FOG episodes or are these differences representative of PD in general? Similarly, how do authors expect that being in the non-optimal OFF state may affect the results? Having appropriate control groups is critical to answering these questions. This is listed as a limitation, but this limitation is much bigger than it was given credit for. Given the likely difficulty to collect more data with many countries being shut down to due COVID-19, this reviewer suggests at the minimum to compare the current results with results in existing literature. Without this comparison, the results that PD-FOG results in the presented ROM differences is not supported.

Response: We appreciate the reviewer’s concern and we made modifications across the manuscript to answer these questions. 

Individuals with FOG are more likely to present a worse gait performance (e.g. asymmetrical steps) than individuals without FOG (Bekkers et al. 2017). A recent study published by our research group brings kinematic gait values of individuals with PD without FOG (Cabeleira, et al. 2019). If kinematic gait parameters are compared with data presented in the current manuscript, it is possible to note that individuals with FOG have greater gait impairments. Even if participants have not presented FOG episodes during gait analysis in the current study, we believe gait would be worse when compared with individuals that never experienced a FOG episode. Thus, we think that these abnormalities on gait would remain – maybe in a lesser extent - even when they are walking without FOG episodes (Bekkers, et al. 2017). 

We agree that the non-optimal OFF state could influence the results. However, when we set this methodology, we aimed two things: a) to avoid the effect of levodopa medication on those cases in which FOG is sensitive to medication (Nonnekes, et al, 2015); b) to focus on the most common and disabling situation for individuals with PD – that is the wearing or partially off state. Once receiving medication, individuals with PD rarely stop to intake it. However, they frequently face difficulties managing the wearing off phase. 

We agree the lack of FOG episodes during gait tests and the non-optimal off medication state are important limitations and we mentioned it in the limitations section (page 16). 

The main objective of this study was to compare both gait conditions (single and dual task) in people with PD and FOG. In this sense, we compared our current results with another research (Ribeiro et al. 2018). We have found only one study evaluating lower limb RoM during dual-task gait in the on state of medication. This study included individuals with PD without FOG. Authors did not find differences between single and dual-task conditions. We discussed our results comparing them with those from Ribeiro et al. 2018. Please see Page 14. 

Bekkers EMJ, Hoogkamer W, Bengevoord A, Heremans E, Verschueren SMP, et al. (2017) Freezing-related perception deficits of asymmetrical walking in Parkinson's disease. Neuroscience 364: 122-129.

Ribeiro T, Sousa AVCd, Lucena LCd, Santiago Ana LMM, Lindquist RR (2018) Does dual task walking affect gait symmetry in individuals with Parkinson’s disease? European Journal of Physiotherapy 21: 8-14.

Nonnekes J, Snijders AH, Nutt JG, Deuschl G, Giladi N, et al. (2015) Freezing of gait: a practical approach to management. Lancet Neurol 14: 768-778.

Cabeleira MEP, Pagnussat AS, do Pinho AS, Asquidamini ACD, Freire AB, et al. (2019) Impairments in gait kinematics and postural control may not correlate with dopamine transporter depletion in individuals with mild to moderate Parkinson's disease. Eur J Neurosci 49: 1640-1648.

Minor Comments:

Methods, Line 105: Why was the age range for inclusion criteria 50-85 years? There is no rationale give for this.

Response: We set this age range in order to include as many individuals with PD as we could. A significant number of people with idiopathic Parkinson's Disease develop symptoms at 50 years of age or older (Pagano, et al. 2016). We recognize the age range could be considered a limitation once we included adults and elderly people. However, our sample was composed of participants with more than 60 years old, as shown in table 1. We included this information in the Methods section (please see page 4):

“age between 50 and 85 years (in order to encompass most of PD population)”

Pagano G, Ferrara N, Brooks DJ, Pavese N (2016) Age at onset and Parkinson disease phenotype. Neurology 86: 1400-1407.

Methods, Lines 111-119: What is the rationale for testing in the OFF state? Or rather, close to OFF state as possible. This is in the Discussion section, but not motivated in either the Intro or Discussion.

Response: We chose to evaluate participants in the off phase in order to avoid dopamine effects on FOG (Nonnekes et al, 2015). However, a drug withdrawal of 12 hours is quite unusual to happen in their real life and a partially off might be more recurrent during their daily living. As we mentioned previously, we aimed: a) to avoid the effect of levodopa medication on those cases in which FOG is sensitive to medication (Nonnekes 2015); b) to focus on the most common daily living situation for individuals with PD – partially off state. Based on your suggestion, we justify this topic rewriting the introduction (page 4), methods (pages 4-6) and discussion (page 13) sections as follows: 

Introduction “…during partially off state of antiparkinsonian medication.”

Methods “We chose to evaluate participants in the off phase in order to avoid dopamine effects on FOG [23]. However, participants were evaluated in the end-of dose of medication (partially or close to OFF state), when levodopa is losing its effect (“wearing off”). Most of the participants did not tolerate withdrawal from levodopa medication for 12 hours in their daily living. For this reason, the OFF-medication phase was defined according to the end-of dose of medication as the participant’s drug regimen in order to reproduce their daily routine where medication is losing its effect. Individuals were instructed to not intake the next dose at the evaluation time, according to each prescribed time of medication intake (it ranged between 4 to 12 hours, according to each participant). If the researchers noticed that individuals were still in “on-phase”, they waited until a subjective “off state” to start the tests.”

Discussion “Additionally, participants performed their walking trials close to their off-antiparkinsonian medication phase (partially off), because this condition may aggravate gait problems, such as FOG [2,8], and it is more common to happen during their daily living.”

Nonnekes J, Snijders AH, Nutt JG, Deuschl G, Giladi N, et al. (2015) Freezing of gait: a practical approach to management. Lancet Neurol 14: 768-778.

Methods, Lines 148-151: Why were gait phases based on percentages instead of gait events? How consistent were these percentages across subjects (with gait events based on marker data)?

Response: We agree with the reviewer point and change gait phases to gait events. Those percentages were estimated according to Hughes, et al. 1979 and we added the word “approximately” before each percentage estimation. More than that, we made modifications across the entire document and on the methods section, as detailed below (page 6):

“In order to better translate for clinical practice, we matched each percentage of the gait cycle based on well-established gait events, as follows: initial contact or heel strike (approx. 0%), load response (approx. 10%), heel off (approx. 30%), opposite initial contact (approx. 50%), toe off (approx. 60%), feet adjacent (approx. 73%), tibial vertical (approx. 87%), and next initial contact or heel strike (approx. 100%) [26].” 

26. Hughes J, Jacobs N (1979) Normal human locomotion. Prosthetics and Orthotics International 3: 4-12.

Methods / Results, Table 2: Only DTE on the gait task was evaluated. What was the Cognitive DTE.

Response: The Dual Task Effect (DTE) was used to verify the speed change between single and dual task gait. DTE was calculated using the formula [DTE = (dual task gait speed – single task gait speed) / single task gait speed x 100%] [30]. Negative speed DTE values indicate a decrement under dual compared to single task. We included this information in methods session as follows (page 6):

 “Also, the gait speed change between single and dual task gait was calculated using Dual Task Effect (DTE) formula [DTE (%) = (dual task gait speed – single task gait speed) / single task gait speed x 100%] [30]. Negative speed DTE values indicate a decrement under dual compared to single task.”

30. Northeast L, Gautrey CN, Bottoms L, Hughes G, Mitchell ACS, et al. (2018) Full gait cycle analysis of lower limb and trunk kinematics and muscle activations during walking in participants with and without ankle instability. Gait Posture 64: 114-118.

The cognitive test used was the word-color interference test while walking. We provided more details about the cognitive test, as follows (page 5):

“Then, they were asked to walk while performing a cognitive test at the same time (dual task gait). The cognitive test performed was the word-color interference test [25] which consists of reading aloud the name of colors written in non-congruent colors.”

25. Stroop JR (1992) Studies of Interference in Serial Verbal Reactions. Journal of Experimental Psychology 121: 15-23.

---

## [Decision Letter · Decision Letter 1]

4 Sep 2020

PONE-D-20-03253R1

Dual-task walking reduces lower limb range of motion in individuals with Parkinson’s Disease and Freezing of Gait. But does it happen during what events through the gait cycle?

PLOS ONE

Dear Dr. Pagnussat,

Thank you for submitting your manuscript to PLOS ONE. After careful consideration, we feel that it has merit but does not fully meet PLOS ONE’s publication criteria as it currently stands. Therefore, we invite you to submit a revised version of the manuscript that addresses the points raised during the review process.

In particular, the reviewer has raised the potential that the results are substantially confounded by differences in walking speed. This is a *significant limitation* that should be noted and addressed clearly within the discussion as it limits the conclusions substantially.

We look forward to receiving your revised manuscript.

Kind regards,

J. Lucas McKay, Ph.D., M.S.C.R.

Academic Editor

PLOS ONE

Reviewers' comments:

Reviewer's Responses to Questions

**Comments to the Author**

1. If the authors have adequately addressed your comments raised in a previous round of review and you feel that this manuscript is now acceptable for publication, you may indicate that here to bypass the “Comments to the Author” section, enter your conflict of interest statement in the “Confidential to Editor” section, and submit your "Accept" recommendation.

Reviewer #1: (No Response)

2. Is the manuscript technically sound, and do the data support the conclusions?

Reviewer #1: Partly

3. Has the statistical analysis been performed appropriately and rigorously? 

Reviewer #1: Yes

4. Have the authors made all data underlying the findings in their manuscript fully available?

Reviewer #1: (No Response)

5. Is the manuscript presented in an intelligible fashion and written in standard English?

Reviewer #1: Yes

6. Review Comments to the Author

Reviewer #1: This study investigates sagittal-plane joint range of motion (RoM) and kinematics during dual-task gait in individuals with Parkinson’s disease (PD) that have freezing of gait. Although the revised manuscript is overall responsive to the previous critiques, one major concern still remains: how much of the identified differences in RoM and kinematics are simply a result of walking at a slower speed?

A significant portion of the Discussion section is focused on the mechanical implications of individual joint kinematic differences. However, there is no discussion of how these differences may simply be due to walking at a slower speed. The subject cohort walked significantly slower in the dual-task condition. The authors also identified some differences in the dual task compared to single task conditions in both joint RoM and kinematics. The reduction in walking speed was significantly correlated with reduced RoM in both conditions. It may be that these individuals reduced their walking speed (e.g., to increase stability?) and as a result their kinematics differed. If this is true, statements such as “lack of ankle plantarflexion would be the major problem related to gait impairments in the swing periods of dual-task gait” would not be true. To tease out how much of these kinematic differences are speed related would require individuals to walk at matched speeds between conditions. The Discussion section should be revised in light of this important point.

Minor Comments:

Line numbers refer to the redline version of the manuscript.

Line 80-81: The authors state that a previous study examined RoM in people with PD without FOG - what did they find?

Line 85-86: The authors state that using discrete parameter statistics it is impossible to visualize movement trajectories into the specific gait phase. “Impossible” is too strong of a statement, as this could be carefully done by analyzing average or change in kinematics within specific sub-phases of gait.

Line 90: Missing a word - “pattern and location of joint kinematic <differences> during a whole movement cycle”

Line 331: replace “exact gait events” with “exact gait phases”</differences>

7. PLOS authors have the option to publish the peer review history of their article (what does this mean?). If published, this will include your full peer review and any attached files.

Reviewer #1: No

---

## [Author Response · Author response to Decision Letter 1]

25 Oct 2020

Review Comments to the Author:

Reviewer #1: This study investigates sagittal-plane joint range of motion (RoM) and kinematics during dual-task gait in individuals with Parkinson’s disease (PD) that have freezing of gait. Although the revised manuscript is overall responsive to the previous critiques, one major concern still remains: how much of the identified differences in RoM and kinematics are simply a result of walking at a slower speed?

R.: We’d thank the reviewer’s contributions to our study. This point is quite relevant. Actually, anyone walking in dual-task will face some degree of gait speed reduction. In order to answer this question in a more precise way, we performed a multiple linear regression analysis. With this analysis, we investigated the contribution of gait speed on the range of motion in each condition separately (Cohen & Cohen, 1983). Our results showed that gait speed was able to explain 70.8% and 49.9% of the hips RoM during single and dual-task gait, respectively, 68.2% and 61.2% of the knees, and 28% and 36.8% of ankles. These results show that gait speed has an important influence on RoM, mainly in hips and knees. We made several modifications across the paper to include and discuss this point. Please see methods (page 7), results and table 3 (pages 9 and 11).

Ref.: Cohen, J. & Cohen, P. (1983). Applied multiple regression/correlation analysis for the behavioral sciences (2nd ed.). Hillsdale, NJ: Lawrence Erlbaum Associates. View

A significant portion of the Discussion section is focused on the mechanical implications of individual joint kinematic differences. However, there is no discussion of how these differences may simply be due to walking at a slower speed. The subject cohort walked significantly slower in the dual-task condition. The authors also identified some differences in the dual task compared to single task conditions in both joint RoM and kinematics. The reduction in walking speed was significantly correlated with reduced RoM in both conditions. It may be that these individuals reduced their walking speed (e.g., to increase stability?) and as a result their kinematics differed. If this is true, statements such as “lack of ankle plantarflexion would be the major problem related to gait impairments in the swing periods of dual-task gait” would not be true. To tease out how much of these kinematic differences are speed related would require individuals to walk at matched speeds between conditions. The Discussion section should be revised in light of this important point.

R.: We agree with the reviewer’s comment. In fact, gait speed may have an influence on RoM when individuals change their attention to another task (i.e. dual-task) and seek for stability. In order to attend this relevant discussion, we performed a regression analysis and included this analysis through the manuscript as detailed above. Moreover, we modified our discussion based on this new idea. The statement that the reviewer mentioned was excluded (page 15) and new points were added regarding the gait speed contribution on lower limb RoM, as well new references about this topic. Please see detailed below:

“Moreover, previous studies discuss an important influence by gait speed on lower limb kinematics [36], concluding that gait speed is angle dependant and dependant on the phase of the gait cycle [37]. It means that the RoM of lower limb joints increase with a faster walk [36]. In order to assess the contribution of gait speed on lower limb RoM in our results we performed a regression analysis, although the best way to inform the speed contribution would be to require individuals to walk at matched speeds between conditions. It seems that a slower or faster walking speed has a strong relation on how much individuals increase the RoM of lower limbs, especially for hips and knees. We hypothesize that when individual focus on another activity (dual-task gait), they seem to reduce the gait speed in order to increase stability and as a result the RoM of lower limb joints decrease.”

Ref.:

36. Han Y, Wang X (2011) The biomechanical study of lower limb during human walking. Science China Technological Sciences 54: 983-991.

37. Hanlon M, Anderson R (2006) Prediction methods to account for the effect of gait speed on lower limb angular kinematics. Gait Posture 24: 280-287.

Minor Comments:

Line 80-81: The authors state that a previous study examined RoM in people with PD without FOG - what did they find?

R.: We added the results about the study mentioned. Please see the modifications bellow:

“A previous study evaluated lower limb RoM (sagittal plane) on people with PD without FOG and control subjects during a single gait [16]. Individuals with PD showed a reduced RoM of hips, knees, and ankles (3 to 5 degrees less, approximately) during a walking performance when compared to control subjects [16].”

Line 85-86: The authors state that using discrete parameter statistics it is impossible to visualize movement trajectories into the specific gait phase. “Impossible” is too strong of a statement, as this could be carefully done by analyzing average or change in kinematics within specific sub-phases of gait.

R.: We agree with the reviewer statement. We rewrite this sentence as follows:

“Traditional gait analysis generally quantifies lower limb RoM as the average of the entire gait cycle through discrete parameter statistics, which is challenging to visualize movement trajectories into specific gait phases [2,14,17].”

Line 90: Missing a word - “pattern and location of joint kinematic during a whole movement cycle”

R.: Done as suggested. 

“SPM is a method able to analyze one-dimensional (1D) data and identify the exact pattern and location of joint kinematics during the entire gait cycle.”

Line 331: replace “exact gait events” with “exact gait phases”

R.: Done as suggested.

---

## [Editor Report · Decision Letter 2]

28 Oct 2020

PONE-D-20-03253R2

Dual-task walking reduces lower limb range of motion in individuals with Parkinson’s Disease and Freezing of Gait. But does it happen during what events through the gait cycle?

PLOS ONE

Dear Dr. Pagnussat,

Thank you for submitting your manuscript to PLOS ONE. After careful consideration, we feel that it has merit but does not fully meet PLOS ONE’s publication criteria as it currently stands. Therefore, we invite you to submit a revised version of the manuscript that addresses the points raised during the review process.

Dear authors, Thank you for taking the time to respond to the reviewer's critiques. While they have been generally satisfied, there are two changes that are required to meet the criteria for publication. First, I encourage you to please more fully address the reviewer's concern regarding the significant limitation in the interpretation of these results due to confounding between the independent variable of task (dual vs. single) and gait speed (slower vs. faster). The inclusion of the regression modeling provides insight to this relationship, but in the context of this experimental design, where task was controlled but gait speed was not, the potential remains for confounding between these two interacting variables. Therefore, please add a clear paragraph to the discussion similar to the following. "One important limitation is that although we demonstrated changes in lower limb kinematics during dual-task gait when compared to single gait, particularly during toe off and heel strike, reductions in gait speed were also observed in the dual-task condition, which raises the potential for confounding. It is unknown whether these changes in lower limb kinematics result directly and uniquely from the dual-task condition, or whether similar changes would be observed in a single-task condition with slower gait speed." Additionally, in multiple places the authors refer to the study as "a cross sectional study." Because the primary independent variable is task condition within a subject (single dual-task vs. dual-task), rather than some other factor across subjects (say, FOG vs. NO FOG), it would be more precise to refer to the study as an "observational within-subjects study." Please change this language in the Abstract, Ethics Statement, and Methods.

We look forward to receiving your revised manuscript.

Kind regards,

J. Lucas McKay, Ph.D., M.S.C.R.

Academic Editor

PLOS ONE

Additional Editor Comments (if provided):

Dear authors,

Thank you for taking the time to respond to the reviewer's critiques. While they have been generally satisfied, there are two changes that are required to meet the criteria for publication.

First, I encourage you to please more fully address the reviewer's concern regarding the significant limitation in the interpretation of these results due to confounding between the independent variable of task (dual vs. single) and gait speed (slower vs. faster). The inclusion of the regression modeling provides insight to this relationship, but in the context of this experimental design, where task was controlled but gait speed was not, the potential remains for confounding between these two interacting variables. Therefore, please add a clear paragraph to the discussion similar to the following.

"One important limitation is that although we demonstrated changes in lower limb kinematics during dual-task gait when compared to single gait, particularly during toe off and heel strike, reductions in gait speed were also observed in the dual-task condition, which raises the potential for confounding. It is unknown whether these changes in lower limb kinematics result directly and uniquely from the dual-task condition, or whether similar changes would be observed in a single-task condition with slower gait speed."

Additionally, in multiple places the authors refer to the study as "a cross sectional study." Because the primary independent variable is task condition within a subject (single dual-task vs. dual-task), rather than some other factor across subjects (say, FOG vs. NO FOG), it would be more precise to refer to the study as an "observational within-subjects study." Please change this language in the Abstract, Ethics Statement, and Methods.

---

## [Author Response · Author response to Decision Letter 2]

4 Nov 2020

Additional Editor Comments (if provided):

Dear authors,

Thank you for taking the time to respond to the reviewer's critiques. While they have been generally satisfied, there are two changes that are required to meet the criteria for publication.

First, I encourage you to please more fully address the reviewer's concern regarding the significant limitation in the interpretation of these results due to confounding between the independent variable of task (dual vs. single) and gait speed (slower vs. faster). The inclusion of the regression modeling provides insight to this relationship, but in the context of this experimental design, where task was controlled but gait speed was not, the potential remains for confounding between these two interacting variables. Therefore, please add a clear paragraph to the discussion similar to the following.

"One important limitation is that although we demonstrated changes in lower limb kinematics during dual-task gait when compared to single gait, particularly during toe off and heel strike, reductions in gait speed were also observed in the dual-task condition, which raises the potential for confounding. It is unknown whether these changes in lower limb kinematics result directly and uniquely from the dual-task condition, or whether similar changes would be observed in a single-task condition with slower gait speed."

Additionally, in multiple places the authors refer to the study as "a cross sectional study." Because the primary independent variable is task condition within a subject (single dual-task vs. dual-task), rather than some other factor across subjects (say, FOG vs. NO FOG), it would be more precise to refer to the study as an "observational within-subjects study." Please change this language in the Abstract, Ethics Statement, and Methods. 

R.: We would like to thank the editor for his contributions to our manuscript. Indeed, to find out how much lower limb kinematics are speed-related, individuals should walk at matched speeds in dual and single-task conditions. The gait speed was not controlled in our study and it might be a potential for confounding. We highlighted this limitation in the discussion section (pages 16-17). We apologize for missing this important critique. Moreover, we changed the study design to "observational within-subjects study" in the Abstract, Ethics Statement, and Methods.

---

## [Editor Report · Decision Letter 3]

17 Nov 2020

Dual-task walking reduces lower limb range of motion in individuals with Parkinson’s Disease and Freezing of Gait. But does it happen during what events through the gait cycle?

PONE-D-20-03253R3

Dear Dr. Pagnussat,

We’re pleased to inform you that your manuscript has been judged scientifically suitable for publication and will be formally accepted for publication once it meets all outstanding technical requirements.

Kind regards,

J. Lucas McKay, Ph.D., M.S.C.R.

Academic Editor

PLOS ONE
---

## [Editor Report · Acceptance letter]

23 Nov 2020

PONE-D-20-03253R3 

Dual-task walking reduces lower limb range of motion in individuals with Parkinson’s Disease and Freezing of Gait. But does it happen during what events through the gait cycle? 

Dear Dr. Pagnussat:

I'm pleased to inform you that your manuscript has been deemed suitable for publication in PLOS ONE. Congratulations! Your manuscript is now with our production department. 

Kind regards, 

on behalf of

Dr. J. Lucas McKay 

Academic Editor

PLOS ONE